# Natural Enemies and Biological Control of Stink Bugs (Hemiptera: Heteroptera) in North America

**DOI:** 10.3390/insects13100932

**Published:** 2022-10-14

**Authors:** Blessing Ademokoya, Kacie Athey, John Ruberson

**Affiliations:** 1Department of Entomology and Nematology, West Florida Research and Education Center, University of Florida, Jay, FL 32565, USA; 2Department of Crop Sciences, University of Illinois at Urbana-Champaign, Urbana, IL 61801, USA; 3Department of Entomology, University of Nebraska, Lincoln, NE 68583, USA

**Keywords:** predators, parasitoids, pathogens, importation biological control, augmentative biological control, conservation biological control, area-wide management, invasive species

## Abstract

**Simple Summary:**

Stink bugs are important pests of many crops in the US, including row crops, vegetables, and tree fruits and nuts. Most stink bug management relies on broad-spectrum, disruptive insecticides with high human and environmental risks associated with them. These issues and increasing pesticide resistance in stink bugs are forcing pest managers to explore safer and more sustainable options. Here, we review the natural enemies of stink bugs in the US, noting that the egg and the late nymphal and adult stages of stink bugs are the most commonly attacked by parasitoids, whereas eggs and young nymphs are the stages most commonly attacked by predators. The effectiveness of stink bugs’ natural enemies varies widely with stink bug species and habitats, influencing the biological control of stink bugs across crops. Historically, biological control of stink bugs has focused on the introduction of exotic natural enemies against exotic stink bugs. Conservation and augmentation methods of biological control have received less attention in the US, although there may be good opportunities to utilize these approaches. We identify some considerations for the current and future use of biological control for stink bugs, including the potential for area-wide management approaches.

**Abstract:**

Stink bugs comprise a significant and costly pest complex for numerous crops in the US, including row crops, vegetables, and tree fruits and nuts. Most management relies on the use of broad-spectrum and disruptive insecticides with high human and environmental risks associated with them. Growing concerns about pesticide resistance in stink bugs are forcing pest managers to explore safer and more sustainable options. Here, we review the diverse suite of natural enemies of stink bugs in the US, noting that the egg and the late nymphal and adult stages of stink bugs are the most commonly attacked by parasitoids, whereas eggs and young nymphs are the stages most commonly attacked by predators. The effectiveness of stink bugs’ natural enemies varies widely with stink bug species and habitats, influencing the biological control of stink bugs across crops. Historically, biological control of stink bugs has focused on introduction of exotic natural enemies against exotic stink bugs. Conservation and augmentation methods of biological control have received less attention in the US, although there may be good opportunities to utilize these approaches. We identify some considerations for the current and future use of biological control for stink bugs, including the potential for area-wide management approaches.

## 1. Introduction

### 1.1. Importance of Stink Bugs as Pests and Challenges in Their Management

Stink bugs are historically important agricultural pests in the US, especially in the southeastern region. They are polyphagous and feed on a wide range of economic crops, including fruits, nuts, vegetables, and grains [1,2,3,4]. Adults and nymphs suck fluids from various plant parts, such as stems, petioles, leaves, flowers, fruits, and seeds. Stink bugs’ feeding activities cause loss of turgidity, stunted growth, delayed maturation, dimpling of fruits, abortion of seeds and fruiting bodies, and shriveled and undersized seeds [1,2,3]. They cause yield loss by attacking marketable plant parts, e.g., kernel damage and ear deformation in maize [5,6,7]; shriveled, discolored, deformed seeds and sometimes complete loss of pods in soybeans [2,3,8]; and boll abortion, reduced lint production, and reduced lint quality in cotton [9,10,11]. While punctures left by their feeding activities can sometimes predispose plants to attack by pathogens, some stink bug species—such as the green stink bug *Chinavia hilaris* (Say), southern green stink bug *Nezara viridula* (L.), red-shouldered stink bug *Thyanta custator* McAtee, and several *Euschistus* species—can act as a vector for phytopathogenic organisms [12,13,14,15]. Annual crop losses in the US, including control costs, are estimated at millions of dollars [2,16,17,18,19,20]. During the growing season, phytophagous stink bug species move between cultivated and non-cultivated habitats (i.e., wild hosts) and from one crop to another, with wild hosts serving as sources/sinks for pest buildup before dispersal into cultivated crops [16,17,18,19,20,21,22,23,24]. Egg laying is the first determinant in stink bug dispersal patterns. Stink bugs lay eggs in masses, and first-instar nymphs usually aggregate in clusters [4,25,26,27]. Movement of older nymphs and adults is influenced by the availability of food and the developmental stage of plants [18,20,26,28,29], as well as host suitability [17,30,31], host preference [32], and host proximity [18,33]. Long-range adult dispersal is aided by strong flight abilities, weather fronts such as winds and hurricanes, and human commercial activities [34,35,36]. This ability to disperse across long geographic ranges is evident from the origin of several species and their subsequent occurrences in many continents, e.g., *N. viridula* [4,37]; the brown marmorated stink bug (BMSB), *Halyomorpha halys* (Stål) [38,39]; the red-banded stink bug, *Piezodorus guildinii* (Westwood) [40]; the African cluster bug, *Agonoscelis puberula* (Stål) [41]; and the Bagrada bug, *Bagrada hilaris* (Burmeister) [42]. As stink bug species cross geographical locations, so too do some of their parasitoids, either through classical biocontrol or fortuitous movement, as is the case with *Trissolcus basalis* (Wollaston) and *Trissolcus japonicus* (Ashmead) [43,44].

Before the eradication of the boll weevil (*Anthonomus grandis* Boheman) and the advent of *Bt*-transgenic maize and cotton targeting lepidopteran pests, such as the cotton bollworm (*Helicoverpa zea* (Boddie)) and the tobacco budworm *Chloridea virescens* (F.) (syn = *Heliothis virescens* (F.)), broad-spectrum insecticides such as pyrethroids that were used to manage these pests indirectly suppressed stink bug populations [45,46,47,48,49,50]. The reduced use of broad-spectrum insecticides within the last two decades following the deployment of *Bt*-transgenic cotton and maize has led to increasing densities of native stink bugs [50,51]. This is in addition to the spread of invasive species such as *P. guildinii* and BMSB, particularly in areas such as the Midwest, where stink bugs were historically not a threat to the major crops in the region [38,52,53,54,55,56,57,58,59,60]. Broad-spectrum insecticides—e.g., carbamates, neonicotinoids, organophosphates, and pyrethroids—are still the most widely used tools for managing stink bugs in the US [7,61,62,63], despite the development of resistance to these insecticides by some stink bug species [51,64,65,66,67,68]. These chemicals vary in efficacy against stink bugs, depending on the method of deployment and the active ingredient [63,69,70], as well as on variations in susceptibility between stink bug species and life stages [66,71,72]. However, negative impacts on beneficial organisms with the tendency to lead to secondary pest outbreaks are a major concern for the use of broad-spectrum insecticides in stink bug management [7,63].

### 1.2. IPM of Stink Bugs in North America and Opportunities for Biological Control

Economically important stink bugs in North America (both native and invasive) present several significant challenges to pest managers [1,2,27]. First, they typically have broad host ranges that encompass both wild and cultivated plants [18]. This attribute allows them to disperse across and reproduce in the landscape in a variety of habitats, from which they can colonize cropping systems. Second, the bugs are generally quite mobile as adults, readily moving between natural areas and managed systems. This mobility across habitat margins tends to express itself in an increased abundance of stink bugs on the edges of crop planting areas relative to the interiors of the fields/orchards—especially early in colonization [73,74,75]. Third, stink bugs feed readily on the fruiting structures of plants [2], directly damaging the economic commodity and thereby imposing lower economic thresholds for management in many crops. Adult stink bugs also tend to move across the landscape in response to the temporal and spatial availability of host plants in reproductive stages. These attributes lend themselves to the implementation of area-wide management approaches to more effectively suppress stink bug populations, in conjunction with local management practices for resident, growing populations within crop systems.

Mass trapping/attract-and-kill methods with pheromones, trap crops (e.g., planting of early-maturing soybeans adjacent to a later-maturing variety), modification of the planting date (usually early planting), and spatially limited insecticide application (e.g., field border sprays) have all shown promise [1,76] but have generally not attained wide adoption by producers in the US, for various reasons [2]. As noted above, broad-spectrum insecticides remain the tools of choice because of the lack of more selective insecticides that effectively target the stink bug complex [1,77]. However, environmental concerns about broad-spectrum insecticides have led to increasing, significant pressure for regulatory reductions in their availability in the US and elsewhere. e.g., [78], and insecticide resistance is a growing concern with stink bugs [65,66]. Challenges of management are more acute for invasive stink bugs that have largely escaped natural controls that can contribute important mortality within and beyond managed systems (see [1] for specific examples).

There are several factors that limit the efficacy of biological control of stink bugs within individual crop fields/orchards. First, broad-spectrum pesticide applications limit the efficacy of biological control in cropping systems once thresholds are attained because of the massive destruction of extant natural enemies by the broad-spectrum insecticides required to suppress pest stink bugs (or other pests). Second, although there are numerous parasitoids and predators, they are quite variable in their occurrence across habitats and their efficacy across stink bug species. e.g., [79,80]. This presents challenges when a complex of stink bugs is of economic importance to the crop, as is often the case. Third, there are relatively few known natural enemies of stink bug nymphs—especially the older instars [2]. This certainly reflects limited study, but may also reflect a relatively invulnerable stage for these pests.

### 1.3. Natural Enemies of Stink Bugs in North America

A large and diverse complex of natural enemies of stink bugs has been identified in North America to date (Table 1, Table 2 and Table 3). This suite of enemies offers considerable opportunity to devise significant contributions to the biological control elements of IPM programs.

#### 1.3.1. Egg Parasitoids

Host use by egg parasitoids involves oviposition in hosts’ eggs, where the parasitoid develops until adult emergence, causing egg mortality and a direct reduction in the pest population prior to crop damage [81]. Egg parasitoids of stink bugs are hymenopterans belonging to the families Platygastridae [82,83,84,85,86], Encyrtidae [87,88], Eupelmidae [89,90,91,92,93], Braconidae [94], and Mymaridae [81] (Table 1). The genera *Telenomus* and *Trissolcus* in the family Platygastridae contain most of the known stink bug egg parasitoids in the US [86,95,96].

Egg parasitoids have evolved multiple adaptations for locating hosts. One of these is the use of feeding- and/or oviposition-induced volatiles [97,98,99,100]. They also make use of an infochemical detour strategy in which chemicals associated with the host’s developmental stages (e.g., nymphs) serve as cues for egg location [101]. For instance, plants infested with nymphs of the neotropical brown stink bug *Euschistus heros* (F.) attracted females of *Telenomus podisi* (Ashmead) [98]. This phenomenon has been studied in other Platygastridae, such as *Trissolcus brochymenae* (Ashmead) and *T. basalis* [102,103]. Egg parasitoids are also known to exploit cues from co-evolved hosts, as when naïve *T. japonicus* were introduced to odor cues from BMSB—their co-evolved host—compared to *Podisus maculiventris* (Say) [104], the wasps responded to BMSB odor cues, but not to those of *P. maculiventris*. *Trissolcus japonicus* will attack *P. maculiventris* eggs, but often do not successfully eclose, leading to a high non-reproductive effect [105]. Egg parasitoids can be generalists or specialists, depending on the number of host taxa utilized. *Telenomus podisi* and *T. basalis*—both generalist parasitoids—have broad diversity in terms of stink bug species attacked, with the former being mainly associated with *Euschistus* species as their primary hosts and the latter most often associated with *N. viridula* [37,84].

Instances of egg parasitoids appearing in the US following the introduction of their hosts from their native ranges are not uncommon. A very recent example is *T. japonicus*—an adventive parasitoid of BMSB that was discovered in the US in 2014 while simultaneously being assessed for possible release in a classical biological control program against the pest [44]. *Trissolcus hyalinipennis* Rajmohana and Narendran—a parasitoid of *Bagrada hilaris* (Burmeister)—is another adventive species that has been discovered in the US [106]. Other discoveries of adventive stink bug egg parasitoids include *Gryon aetherium* Talamas [107], *Psix striaticeps* (Dodd) [108], and *Ooencyrtus nezarae* Ishii [109]. However, it should be noted that *O. nezarae* is a generalist that parasitizes several heteropteran families, including Pentatomidae [109,110,111,112,113,114], in other parts of the world. Among the pentatomids parasitized by *O. nezarae* are *Piezodorus hybneri* (Gmelin), *Eysarcoris guttiger* (Thunberg) [110,112], *N. viridula* [37,115], *Euschistus* sp., *Acrosternum* sp., *Edessa* sp., and *Thyanta* sp. [116]. However, *O. nezarae* in North America have only been reported as recovered from the eggs of the kudzu bug (*Megacopta cribraria* (F.) to date [117].

Egg parasitoids are noted to utilize and benefit from plant floral resources—most notably carbohydrates (e.g., [118,119])—which may allow the parasitoids to be utilized effectively in biological control programs.

**Table 1 insects-13-00932-t001:** Parasitoids of stink bugs reported in North America, with hosts, stages attacked, and coarse distribution.

Order, Family	Species	Host Species ^1^	Stage(s) Attacked ^2^	Distribution	Native	References
Hymenoptera						
Braconidae	*Aridelus rufotestaceus* Tobias	Nv, Es	N	Southeastern US	N	[94,120]
	*Aridelus fisheri* (Viereck)	Es	N?	Eastern US	Y	[87]
Encyrtidae	*Hexacladia hilaris* Burks	Ch, Nv	N, A	Central and Eastern US	Y	[82,88,89]
	*Hexacladia smithi* Ashmead	Ecr, Es	N, A	Southeastern US	Y	[88,89]
	*Ooencyrtus californicus* Girault	Bh	E	Western US		[107,121]
	*Ooencyrtus johnsoni* (Howard)	BMSB, Cs, Es, Mh, Pm	E		Y	[43,79,88,92,122]
	*Ooencyrtus lucidus* Triapitsyn and Ganjisaffar	Bh	E	Western US	Y	[123,124]
	*Ooencyrtus mirus* Triapitsyn & Power	Bh	E	Western US	N	[122]
	*Ooencyrtus nezarae* Ishii		E	Southeastern US	N	[109]
	*Ooencyrtus* sp.	Es, Nv, Pm, Bh	E		Y	[79,83,86,89,90,121,125,126,127,128]
	*Ooencyrtus submetallicus* (Howard)	Nv	E	Southeastern US	Y	[89]
Eupelmidae	*Anastatus mirabilis* (Walsh and Riley)	BMSB, Es	E	Western, Central, and Southeastern US; Mexico	Y	[79,87,90,125,126,129]
	*Anastatus pearsalli* Ashmead	BMSB	E	Western US		[79,129]
	*Anastatus reduvii* (Walsh and Riley)	BMSB, Ch, Es	E	Western, Central, and Southeastern US; Mexico; Canada	Y	[79,87,88,90,91,93,122,125,126,127,128,129,130,131,132]
	*Anastatus* sp.	BMSB, Ch, Pm, Es	E			[79,86,89,92,93]
Mymaridae	Undefined sp.	Mh	E			[81]
Platygastridae	*Gryon obesum* Masner	Es, Pg	E	Southern US	Y	[79,82,89,125,126]
	*Gryon* sp.	Nv, Bh	E	Western US	Y	[89,121]
	*Gryon aetherium* Talamas	Bh	E	Western US		[107,121]
	*Gryon myrmecophilum* Talamas	Bh	E	Eastern US; Mexico		[133]
	*Idris elba* Talamas	Bh	E	Mexico		[134]
	*Psix striaticeps* (Dodd)	Nv, Pg, Cm	E	Southeastern US	N	[108]
	*Telenomus calvus* Johnson	Pm	E	Southeastern US	Y	[84,89,135]
	*Telenomus cristatus* Johnson	Ch	E	Southeastern US	Y	[84]
	*Telenomus persimilis* Ashmead	BMSB	E	Eastern US	Y	[79]
	*Telenomus podisi* (Ashmead)	BMSB, Es, Ev, Ei, Et, Op, Pg, Pm, Nv, Ch, Tc, Bh	E	US; Canada; Mexico	Y	[60,79,82,83,84,85,86,87,89,90,91,92,93,122,125,126,127,130,131,133,135,136]
	*Trissolcus basalis* (Wollaston)	Bh, Ch, Es, Nv, Pm, Tc	E	Southern US; Hawaii; Mexico	N	[43,81,82,84,89,90,125,126,127,133,135,137]
	*Trissolcus brochymenae* (Ashmead)	BMSB, Ch, Es, Mh, Pg, Pm	E	US; Canada; Mexico	Y	[43,79,81,85,87,92,122,125,126,127,130,135,136,138]
	*Trissolcus cosmopeplae* (Gahan)	BMSB, Eco	E	US; Canada	Y	[79,138]
	*Trissolcus cristatus* Johnson	Ch	E	Southeastern US		[84]
	*Trissolcus edessae* Fouts	BMSB, Es, Ch, Pm	E	Eastern US	Y	[79,82,84,89,92,122,125,126,127]
	*Trissolcus erugatus* Johnson	Tc, Bh	E	Western US; Canada	Y	[121,138]
	*Trissolcus euschisti* Ashmead	BMSB, Ch, Es, Et, Pm, Tc	E	US; Canada; Mexico	Y	[79,82,83,84,85,86,92,122,125,127,129,135,136,139]
	*Trissolcus hullensis* (Harrington)	Bh, BMSB, Es, Pm	E	US; Canada; Mexico	Y	[79,122,129,130,138]
	*Trissolcus hyalinipennis* Rajmohana & Narendran	Bh	E	Western US	N	[106]
	*Trissolcus japonicus* (Ashmead)	BMSB, Pm, Ch, Es, Eco	E	US; Canada	N	[44,85,92,105,129,131,140,141,142]
	*Trissolcus occiduus* Johnson	Cs	E	Western US	Y	[130,138]
	*Trissolcus solocis* Johnson	BMSB *, Nv, Pm	E	Southeastern US; Mexico	Y	[89,126,127,137]
	*Trissolcus thyantae* Ashmead	BMSB, Es, Et, Ev, Tc	E	Eastern US; Canada	Y	[79,90,91,135,136,138]
	*Trissolcus utahensis* (Ashmead)	Bh, BMSB, Cs, Ei, Eco, Pm, Tc	E	Western US; Canada; Mexico	Y	[79,83,88,125,138,143]
	Undefined sp.	BMSB	E	Western US		[79]
Diptera						
Sarcophagidae	*Sarcodexia innota* (Walker)	Nv		Southeastern US; Mexico	Y	[144]
Tachinidae	*Beskia aelops* (Walker)	Eic, Nv, Op	A	Southern US; Mexico	Y	[89,145]
	*Cylindromyia armata* Aldrich	Cs		US; Canada; Mexico	Y	[145,146]
	*Cylindromyia binotata* (Bigot)	Ev, Es, Tc	N, A	US; Canada; Mexico	Y	[82,89,147]
	*Cylindromyia euchenor* (Walker)	Cl, Eq, Eo, Es, Ecra Eic, Et, Pm	A	US; Canada; Mexico	Y	[82,89,146,148]
	*Cylindromyia fumipennis* (Bigot)	Ev, Es, Pm	N, A	US; Canada; Mexico	Y	[60,89,147,149]
	*Cylindromyia* sp.	Ev, Cl	N, A		Y	[60,143]
	*Euclytia flava* (Townsend)	Es, Tc	A	US; Canada	Y	[82,89,149]
	*Euthera* sp.	Ev	A		Y	[60,150]
	*Euthera tentatrix* Loew	Es, Eo, Eq, Ev, Eic, Et, Nv, Op, Pg, Pm, Tc	N, A	US; Canada; Mexico	Y	[82,89,143,146,147,149,150,151]
	*Gymnoclytia immaculata* (Macquart)	Es, Ev	A	US; Canada; Mexico	Y	[146,148,152]
	*Gymnoclytia occidentalis* Townsend	Eco	A	US; Canada	Y	[83,146]
	*Gymnoclytia occidua* (Walker)	Es, Ev, Et, Tc	A	US; Canada; Mexico	Y	[82,146,147,148,149,150]
	*Gymnoclytia unicolor* (Brooks)	Es, Tc	A	US		[82,89,149]
	*Gymnosoma filiola* Loew	Eco, Cl	A	US; Canada; Mexico	Y	[83,146]
	*Gymnosoma fuliginosum* (Robineau-Desvoidy)	Eim, Es, Ev, Ch, Cl, Cs	N, A	US; Canada; Mexico	Y	[82,146,148,152,153]
	*Gymnosoma par* (Walker)	Tc	A	Canada; Northern and Eastern US	Y	[82,146]
	*Hemyda aurata* (Robineau-Desvoidy)	Ch, Pm	A	Southeastern US	Y	[89,146,148]
	*Phasia chilensis* (Macquart)	Cl		US; Mexico	Y	[145,146]
	*Phasia* sp.	Bs				[143]
	*Trichopoda lanipes* (F.)	Pm, Nv	A	US; Canada; Mexico	Y	[145,146]
	*Trichopoda pennipes* F.	BMSB, Es, Ev, Mh, Nv, Ch, Pg, Tc	N, A	US; Canada; Mexico	Y	[79,82,89,145,146,148,149,154]
	*Trichopoda* sp.	Cl, Bs				[143]
	Undefined sp.	BMSB, Ch, Es, Et, Ev, Pm, Tc	A	Central US		[155]

^1^ Stink bug species abbreviations: Bh = *Bagrada hilaris* (Burmeister), BMSB = *Halyomorpha halys* (Stål), Bs = *Brochymena*
*sulcata* Van Duzee, Ch = *Chinavia hilaris* (Say), Cl = *Chlorochroa ligata* (Say), Cs = *Chlorochroa sayi* (Stål), Eco = *Euschistus conspersus* Uhler, Ecr = *Euschistus crenator* (F.), Ecras = *Euschistus crassus* Dallas, Eic = *Euschistus ictericus* (L.), Eim = *Euschistus impictiventris* Uhler, Eo = *Euschistus*
*obscurus* (Palisot), Eq = *Euschistus quadrator* Rolston, Es = *Euschistus servus* (Say), Et = *Euschistus tristigmus* (Say), Ev = *Euschistus variolarius* (Palisot), Mh = *Murgantia histrionica* (Hahn), Nv = *Nezara viridula* (L.), Op = *Oebalus*
*pugnax* (F.), Pg = *Piezodorus guildinii* (Westwood), Pm = *Podisus maculiventris* (Say) (predator), Tc = *Thyanta custator* (F.). ^2^ Stink bug stage abbreviations: E = egg, N = nymph, A = adult. * Poor suitability.

#### 1.3.2. Nymphal Parasitoids

As noted by Jones [37] and McPherson and McPherson [2], very few parasitoids are known globally that are focused primarily on nymphal instars of stink bugs (Table 1). This likely reflects a lack of parasitoids of these life stages rather than survey bias, although surveys of stink bug nymphs for parasitism are rare in the published literature relative to surveys of eggs and adults.

Buschmann and Whitcomb [89] reared a single specimen of the gregarious encyrtid *Hexacladia hilaris* Burks from a nymph of *N. viridula* in Florida. Jones [37] noted that *H. hilaris* has been reared from stink bugs a few times in North America, from two hosts: *N. viridula* and *C. hilaris*. However, it seems to be primarily an adult parasitoid. This parasitoid was recently found in Brazil, where it was reared from adults of *Chinavia erythrocnemis* (Berg) [156]. *Hexacladia hilaris* has been rarely encountered on pest stink bugs in North America, based on documented nymphal and adult surveys (e.g., [89,94,120].

The exotic, solitary braconid endoparasitoid *Aridelus rufotestaceus* Tobias (originally described in eastern European Georgia by Tobias [157]) was reported in Italy in 1998 and 1999 by Shaw et al. [158]. This parasitoid was subsequently reared from nymphs of *N. viridula* collected in Georgia, US, in 2007 [94], and later in New Zealand [159]. In the US, the parasitoid was reared from nymphs of *N. viridula* and nymphs and adults of *Euschistus servus* (Say) collected in surveys of soybeans in southern Georgia, US, in 2008 and 2009 [94]. This parasitoid is clearly established in the Southern US, but the parasitism rates of both stink bug species by this parasitoid were below 1% of all stink bug nymphs collected in surveys. How long it has been present in the Americas is unknown. At present, its impact on stink bug populations appears to be negligible. *Aridelus rufotestaceus* is a thelytokous parasitoid that prefers attacking earlier stink bug instars (2nd–4th instars), and these instars appear to be most suitable for reproduction [158]. It will not readily sting adult stink bugs. It can occasionally emerge from adult hosts after parasitizing nymphs, but typically completes its development in the nymphal stages. Its longevity and reproduction are enhanced by the availability of carbohydrate sources [160].

#### 1.3.3. Adult Parasitoids

There is a diverse complex of adult parasitoids of the North American stink bug complex (Table 1), although at least a few of them are also known to attack late-stage nymphs as well as adult stink bugs. This complex is heavily dominated by tachinid flies, with some encyrtid wasps (Table 1). Of these, *Trichopoda pennipes* F. is the most frequently encountered and widely studied tachinid parasitoid of stink bugs in North America. This solitary endoparasitoid oviposits external eggs on the stink bug cuticle, and the eggs can be found virtually anywhere on the host’s body (although preference for the thoracic venter is documented [161]). Superparasitism of hosts can be quite high, but only a single parasitoid will emerge successfully [161]. Limited superparasitism can enhance the likelihood of successful parasitism, but higher superparasitism becomes detrimental [161]. Parasitism of stink bugs by tachinids can reach high levels but is usually low–moderate, and adult parasitoids’ impact on stink bug populations and crop damage is likely limited as the parasitoids are koinobionts, i.e., parasitized adult stink bugs continue feeding and reproducing [162,163] unless parasitized shortly after adult emergence of the stink bug, which essentially shuts down reproduction [162]. However, for *T. pennipes* and *Trichopoda giacomellii* (Blanchard) at least, the longevity and fecundity of parasitized hosts are reduced significantly [161,162,163]. Presumably this is true for other adult parasitoids as well. Furthermore, male stink bugs of *N. viridula* tend to be disproportionately parasitized by tachinids due to their release of an aggregation pheromone that is attractive to the parasitoids [164,165,166] (see also [154,167]). *Trichopoda* species are known to utilize floral resources [125,132,168], creating opportunities for manipulating them in the field (see below Table 2).

**Table 2 insects-13-00932-t002:** Predators of stink bugs reported in North America, with hosts, stages attacked, and coarse distributions.

Order, Family	Species	Prey Species ^1^	Stage(s) Attacked ^2^	Distribution	Native	References
Chiroptera						
Vespertilionidae	*Eptesicus fuscus* (Beauvois)	BMSB, Ch		North America	Y	[169,170,171]
	Undefined sp.	Nv				[172]
Rodentia						
Muridae	*Mus musculus* L.	Nv	E	North America	Y	[96,173]
Araneae						
Agelenidae	Undefined sp.	BMSB	A			[174] ^5^
Araneidae	Undefined sp.	BMSB	A			[174] ^5^
Oxyopidae	*Oxyopes* sp.	BMSB, Nv, Ch	E, N		Y	[96,175,176]
	*Oxyopes salticus* (Hentz)	BMSB, Ch, Es, Et, Nv, Pg, Eq, Tc	E, N	US; Canada	Y	[177,178,179] [176] ^3^
	*Peucetia viridans* (Hentz)	Es		Southern US; Mexico	Y	[179] ^5^
Pholcidae	Undefined sp.	BMSB	A			[174] ^5^
Tetragnathidae	*Tetragnatha* sp.	Nv	N			[96]
	Undefined sp.	BMSB	E			[180] ^3^
Theridiidae	Undefined sp.	BMSB	A			[174] ^5^
Thomisidae	*Misumenops* sp.	Nv	N			[96]
	*Mecaphesa asperata* (Hentz)	Nv, Es, Pg		North America	Y	[179]
Trachelidae	*Trachelas* sp.	BMSB	E			[181]
Lycosidae	*Pardosa* sp.	Nv	N			[96]
Salticidae	*Phidippus audax* (Hentz)	Nv	N	US; Mexico	Y	[177,182]
	Undefined sp.	BMSB	E			[180] ^3^
Uloboridae	Undefined sp.	BMSB	A			[174] ^5^
Opiliones						
Phalangiidae	Undefined sp.	Nv	N			[96]
Isopoda						
Armadillidiidae	*Armadillidium vulgare* (Latreille)	Nv	E	North America	N	[96]
Blattodea						
Blattidae	*Blatta orientalis* L.	BMSB	E	North America	N	[181]
Coleoptera						
Anthicidae	*Anthicus cervinus* Laf.	Nv	N	North America	Y	[182]
	*Notoxus calcaratus* Horn		N	North America	Y	[96]
	*Notoxus monodon* (F.)	Nv		North America	Y	
Cantharidae	Undefined sp.	BMSB	E			[180] ^3^
Carabidae	*Harpalus* spp.	BMSB	E			[180] ^3^
	*Laemostenus complanatus* (Dejean)	BMSB	E		N	[181,183]
	*Lebia analis* Dejean	Nv	E			[177,182]
Coccinellidae	*Coccinella californica* (Mannerheim)		N	Western US	Y	[96]
	*Coccinella novemnotata* Herbst		N	US; Canada	Y	[96]
	*Coccinella septempunctata* (L.)	BMSB, Es	N	North America	N	[96,175], [184] ^3^
	*Coleomegilla maculata* (DeGeer)	BMSB, Nv	E, N	North America	Y	[175,177,182], [180,185] ^3^
	*Cycloneda sanguinea* (L.)	Nv	E, N	Southern US; Mexico	Y	[177]
	*Harmonia axyridis* (Pallas)	BMSB, Es, Nv, Pg,	E, N	North America	N	[180] ^3^ [7,96,179]
	*Hippodamia convergens* Guerin	Ch, Es, Eg, Et Nv, Pg, Tc,	N	North America	Y	[96,178,179]
	*Scymnus* sp.	Es, Eq, Nv, Pg				[179]
Collembola						
Entomobryidae	Undefined sp.	Bh	E			[181]
Dermaptera						
Forficulidae	*Euborellia annulipes* (H. Lucas)	BMSB	E	US; Canada	N	[181]
	*Forficula auricularia* L.	BMSB	E	North America	N	[181]
	Undefined sp.	BMSB	E			[180] ^3^ [7,79]
Hemiptera						
Anthocoridae	*Orius insidiosus* (Say)	BMSB, Ch, Es, Et, Nv, Pg, Tc,	E	North America	Y	[186] ^4^ [177,178,179] [180] ^3^
	*Orius* sp.	BMSB, ES, Nv	E			[7,175]
Geocoridae	*Geocoris atricolor* Montandon	Nv	N	Western US	Y	[96]
	*Geocoris pallens* Stål	Nv	E, N	Western US	Y	[96]
	*Geocoris punctipes* (Say)	Et, Nv, Pg, Tc	E, N	Eastern US	Y	[96,177,178,182,187] [186] ^4^
	*Geocoris uliginosus* (Say)	Et, Nv, Pg	E	Eastern and Central US	Y	[178,182,187], [186] ^4^
	*Geocoris* spp.	BMSB, Ch, Es, Eq, Et, Nv, Pg, Tc,	E, N			[7,175,179]
Miridae	*Pseudatomoscelis seriatus* (Reuter)	Nv	E	North America	Y	[187]
	*Spanagonicus* sp.	Nv	E		Y	[187]
Nabidae	*Nabis americoferus* Carayon	Nv	N	North America	Y	[96]
	*Nabis roseipennis* Reuter	Nv	E, N	North America	Y	[177,182]
	*Nabis* sp.	BMSB, Ch, Nv	N			[175] [184] ^3^
	*Tropiconabis capsiformis* (Germar)	Nv	N	Southern US; Mexico	Y	[182]
Pentatomidae	*Podisus maculiventris* (Say)	Bh, BMSB, Es, Nv, Tc	E. N	US; Canada	Y	[49,177,180,182], [185] ^3^ [186] ^4^
Reduviidae	*Arilus cristatus* (L.)	BMSB	E, N, A	North America	Y	[7,188]
	*Sinea diadema* (F.)	Nv	N	North America	Y	[96]
	*Sinea* sp.	Nv	N		Y	[182]
	*Zelus renardii* Kolenati	Es, Et, Nv	N	US; Mexico	Y	[96,179]
	*Zelus* sp.	Es				[175]
	Undefined sp.	BMSB	E, N			[180], [184] ^3^
Hymenoptera						
Formicidae	*Monomorium ergatogyna* Wheeler	Bh	E	Western US	Y	[181]
	*Solenopsis invicta* Buren	Ch, Es, Eq, Et, Nv, Pg	E	Southern US; Mexico	N	[177,178,179,182,187,189]
	*Solenopsis xyloni* McCook	Bh	E	Southern US; Mexico	Y	[181]
	Undefined sp.	BMSB	E			[122,180]^3^
Crabronidae	*Astata bicolor* Say	BMSB	N	North America	Y	[7]
	*Astata unicolor* (Say)	BMSB	A, N	North America	Y	[7,188,190]
	*Astata* sp.	BMSB	E			[183]
Mantodea						
Mantidae	*Tenodera sinensis* (Saussure)	BMSB	N, A	US; Canada	N	[7]
Neuroptera						
Chrysopidae	*Chrysopa* spp.	BMSB	E			[180] ^3^
	*Chrysoperla carnea* (Stephens)	BMSB	E, N	US; Canada	N	[185] ^3^
	*Chrysoperla rufilabris* (Burmeister)	Nv	E	North America	Y	[187]
	Undefined sp.	BMSB	E, N			[7]
Orthoptera						
Acrididae	*Schistocerca obscura* (F.)	Nv		Eastern and Southern US	Y	[182]
	*Schistocerca* sp. nymph	Nv				[182]
	Undefined sp.	BMSB	E			[184] ^3^
Gryllidae	*Oecanthus* spp.	BMSB	E			[180] ^3^
Tettigoniidae	*Atlanticus testaceus* Scudder	BMSB	E	North America	Y	[180] ^3^
	*Conocephalus fasciatus* (De Geer)	Nv	E	US; Canada	Y	[182]
	*Conocephalus strictus* (Scudder)	BMSB	E	North America	Y	[180] ^3^
	*Neoconocephalus ensiger* (Harris)	BMSB	E	North America	Y	[180] ^3^
	*Neoconocephalus robustus* (Scudder)	BMSB	E	North America	Y	[180] ^3^
	*Neoconocephalus* spp.	BMSB	E			[180] ^3^
	*Orchelimum nigripes* Scudder	Nv	E	North America	Y	[182]
	*Orchelimum* spp.	BMSB	E			[180] ^3^
	*Orchelimum vulgare* (Harris)	BMSB	E	Central and Eastern US	Y	[180] ^3^
	Undefined sp.	Nv, BMSB	E		Y	[187], [180] ^3^

^1^ Stink bug species abbreviations: Bh = *Bagrada hilaris* (Burmeister), Bs = *Brochymena sulcata* Van Duzee, Ch = *Chinavia hilaris* (Say), Cl = *Chlorochroa ligata* (Say), Cs = *Chlorochroa sayi* (Stål), Eco = *Euschistus conspersus* Uhler, Ecr = *Euschistus crenator* (F.), Ecras = *Euschistus crassus* Dallas, Eic = *Euschistus ictericus* (L.), Eim = *Euschistus impictiventris* Uhler, Eo = *Euschistus obscurus* (Palisot), Eq = *Euschistus quadrator* Rolston, Es = *Euschistus servus* (Say), Et = *Euschistus tristigmus* (Say), Ev = *Euschistus variolarius* (Palisot), BMSB = *Halyomorpha halys* (Stål), Mh = *Murgantia histrionica* (Hahn), Nv = *Nezara viridula* (L.), Op = *Oebalus pugnax* (F.), Pg = *Piezodorus guildinii* (Westwood), Pm = *Podisus maculiventris* (Say) (predator), Tc = *Thyanta custator* (F.). ^2^ Stink bug stage abbreviations: E = egg, N = nymph, A = adult. ^3^ Laboratory feeding trials. ^4^ These predators were observed feeding on stink bug eggs and nymphs, but the identity of the stink bug was not mentioned. The author added Es and Nv egg masses to the peanut field. ^5^ Spiders in dwellings eating overwintering BMSBs.

#### 1.3.4. Predators

Sentinel egg mass studies are very common for the study of the biological control of stink bugs (e.g., many studies noted in [79] for BMSB). Such studies have been used for a long time to identify egg parasitoids that can successfully attack a given stink bug species. These studies are also often used to determine predation rates. The drawback to this approach is that, unless predators are caught in the act, there is no way to identify the actual predators that are doing the damage to the eggs. Many of these studies separate the predators into two groups: sucking predators and chewing predators. Determining the identity and impact of predators of stink bugs is more complicated than simply using sentinel egg masses. In addition, determining predation and parasitism on the mobile life stages is not straightforward.

Stam et al. (1987) [182] utilized radiolabeling (^32^P) to elucidate predators of young stink bug (*N. viridula*) nymphs in soybeans. They identified a diverse suite of predators using this tool, but the complexities and risks of using radiolabeling limit its use and value. An increasingly common method for determining predation on pestiferous insects is molecular gut content analysis (MGCA) [175,191,192,193]. This has been used to investigate the identities of generalist predators consuming stink bugs using DNA (polymerase chain reaction (PCR)) [175,178,179] and proteins (enzyme-linked immunosorbent assay (ELISA)) [177]. While these studies vary considerably in their conclusions about the impacts of generalist predators on stink bugs, they do confirm the identity of several generalist predators consuming stink bugs, including *Orius* sp. and *Geocoris* spp. [175,177,178,179]. These predators have been observed consuming stink bugs in numerous studies [7,96,175,178,179,186,187]. In the case of DNA, which is the more common method for gut content analysis, the life stage of the prey item cannot be determined—only its identity. An additional drawback to this method is the inability to quantify predation. With sentinel egg mass studies, we can estimate the impact of predators as a group on stink bugs. We can compare the impact of chewing predators to that of sucking predators. In the case of parasitoids, we can even obtain the identity of the natural enemy. Using MGCA, we can determine the identity of the predator, but we cannot quantify its impact. Scavenging and secondary predation cannot be separated from primary predation using this method either [194]. Both observations and MGCA have been used to determine predators of a variety of stink bug species.

The BMSB has been studied frequently (see the review in [79]). Predation on BMSB adults and nymphs by provisioning wasps specializing in stink bugs—e.g., *Astata bicolor* Say [7], *Astata unicolor* (Say) [7,188,190], and *Bicyrtes quadrifasciata* (Say) [7,188]—has been documented. Several studies and reviews have investigated predation on BMSBs in laboratory feeding trials [174,176,185] and sentinel egg mass studies [92,93,105,122,127,130,181,195,196]. These studies separated predation into two categories—chewing predators and sucking predators—based on the egg damage. In a review of sentinel egg mass studies for the BMSB, predation levels on BMSB eggs were generally less than 15% but were sometimes found to be as high as 30% [79]. The trend of chewing predators attacking BMSB eggs at a higher rate than piercing/sucking predators was found in several studies [180]. Predation on eggs was consistently low—7% at the highest. The authors did not identify any of the predators beyond chewing vs. sucking [196]. In another study in an ornamental landscape, predation was very low, accounting for 4.4% of egg mortality [122]. In a large sentinel egg mass study across several states, BMSB egg mortality due to natural enemies was very low (about 20%) and was mostly due to chewing predation [195]. In apple orchards in Minnesota, the authors found that only 2% of egg mortality was attributed to predation [197]. Occasionally, researchers will use photography to catch the predators in the act of consuming stink bug eggs. Using sentinel egg masses, [181] identified several predators that were consuming eggs in the sentinel masses. The main predators detected on the BMSB egg masses were *Laemostenus complanatus* (Dejean) (Coleoptera: Carabidae), the earwigs *Forficula auricularia* L. and *Euborellia annulipes* (Lucas) (Dermaptera: Forficulidae), the cockroach *Blatta orientalis* L. (Blattodea: Blattidae), and a spider, *Trachelas* sp. (Araneae: Trachelidae).

Many studies have investigated BMSB egg predation, but other pestiferous stink bugs also have been studied using sentinel egg masses. Much like the BMSB, the predation is typically characterized as due to chewing or sucking predators, with predator species rarely identified. An early study on *Euschistus* spp. and *C. hilaris* in soybeans and alfalfa showed that chewing predators had a greater impact on egg mortality overall, but sucking predators were more effective in alfalfa in certain years, suggesting that broad generalizations of predator effectiveness may not be applicable [86]. Some of these groups have been tested for the consumption of stink bugs. In a previous study, *F. auricularia* and *B. orientalis* were tested for the consumption of *N. viridula*, *E. servus*, and *C. hilaris*; there were zero gut content positives. It is possible that the decay rate of stink bug DNA in the guts of these predators is very fast, making it difficult to detect predation [198]. Alternatively, the different habitats in which these studies were conducted may have influenced the differences in predation by these species, i.e., cotton and soybeans [175] versus urban trees [181]. For example, the highest rates of predation on sentinel eggs of *N. viridula* in soybeans and peanuts were attributed to tettigoniids and red imported fire ants (*Solenopsis invicta* Buren (Hymenoptera: Formicidae)), respectively, in a replicated large-plot study [181]. Thus, habitat/crop type influences predators’ presence and relative activity.

The Bagrada bug, *Bagrada hilaris* Burmeister (Hemiptera: Pentatomidae), is an invasive pest of cruciferous crops (Brassicaceae). Sentinel egg masses were deployed over a three-year period, and chewing predators were found to be much more common in the field, with piercing/sucking predators rarely found; the predation rates varied between sites and years (11–33%) [121]. The identity of these predators could not be determined. Bagrada bug eggs are deposited in the soil, so they would be expected to have a different predator complex than other pestiferous stink bugs. One observational study using sentinel egg masses and imaging identified several predators consuming Bagrada bugs [181]. Ants were the most common predators of Bagrada bug eggs—mainly *Solenopsis xyloni* McCook and *Monomorium ergatogyna* Dubois (both in Hymenoptera: Formicidae)—followed by collembolans in the family Entomobryidae. Collembola are generally considered to be detritivores, but they were observed consuming Bagrada bug eggs often in this study [181]. Although the Bagrada bug has a different method of egg deposition, this study highlights the need to consider groups that are not usually considered predators when investigating egg predation.

#### 1.3.5. Vertebrate Predators

In addition to invertebrate predation, there is some evidence of vertebrates—particularly mammals—preying on stink bugs, e.g., mice (*Mus musculus* L. (Rodentia: Muridae)) [96,173], big brown bats (*Eptesicus fuscus* (Beauvois) (Chiroptera: Vespertilionidae) [169,170,171]), and an unidentified bat species [172]. The mice were observed preying on eggs [96,173], but the bat predation was based on gut content identification [169,170,171,172].

#### 1.3.6. Pathogens

Few pathogens of stink bugs have been reported in North America (Table 3). This paucity of reports may reflect a limited amount of searching for them rather than a lack of pathogens. However, stink bugs’ access to entomopathogenic bacteria and viruses that require ingestion is likely constrained by the bugs’ piercing/sucking feeding strategy (see [199]). Furthermore, at least some stink bugs produce fungistatic compounds that reduce the effectiveness of entomopathogenic fungi [200,201], which helps account for the low levels of fungal infections reported in stink bug populations [202]. The application of fungicides to many cropping systems may also limit the distribution and efficacy of naturally occurring fungal entomopathogens against stink bugs in production systems. Numerous studies on fungal pathogens of stink bugs have been conducted in South America (focused heavily on *Beauveria bassiana* (Bals.-Criv) Vuill. and *Metarhizium anisopliae* (Metsch.) Sorokin isolates) (both in Hypocreales: Clavicipitaceae), but this has not been the case for North America.

Various protozoans have been reported from the guts of stink bugs in the US (Table 3), and they can occur at high incidences in sampled populations [203,204,205,206]. The role of these microorganisms in stink bug population dynamics is unknown, and it is unclear whether the protozoans adversely affect the biology of the stink bugs [203]. Furthermore, the mechanism of transmission for the trypanosomes has not been elucidated for stink bugs. Vertical transmission appears to be limited; Fuxa et al. [203] found no evidence for transovarial transmission and, instead, proposed transmission through plants, although they detected no trypanosomes in host plants. Transmission may occur through fecal feeding by the bugs, as the trypanosomes are common in the feces of infected bugs [203].

Mermithid nematodes have also been reported from several stink bug species in North America (Table 3). Observations suggest that the incidence of such parasitism in economically important stink bugs is quite low, and their impact on stink bug populations is unknown [207,208].

**Table 3 insects-13-00932-t003:** Pathogens of stink bugs reported in North America, with hosts, stages attacked, and coarse distributions.

Order, Family	Species	Prey Species ^1^	Stage(s) Attacked ^2^	Distribution	Native	References
Microsporidia						
Nosematidae	*Nosema maddoxi* Becnel, Solter, Hajek, Huang, Sanscrainte, & Estep	BMSB, Ch, Es, Et	N, A	US	Y	[209,210]
Hypocreales						
Cordycipitaceae	*Beauveria bassiana* (Bals.-Criv) Vuill.	Op ^3^, Pg	N, A	Southeastern US; Cuba	Y	[211,212,213]
Trypanosomatida						
Trypanosomatidae	Unknown	Nv	N, A	Southeastern US	Y	[203]
	*Blastocrithidia* spp.	Bh	?	Western US	Y	[205]
Nematoda						
Mermithidae	*Hexamermis* sp.	Nv, Pg, Ch	N, A	Southeastern US	Y	[207,208]
	*Agamermis* sp.	Ch, Es, *Euschistus* sp.	N, A	Southeastern US	Y	[208]

^1^ Stink bug species abbreviations: Bh = *Bagrada hilaris* (Burmeister), BMSB = *Halyomorpha halys* (Stål), Ch = *Chinavia hilaris* (Say), Es = *Euschistus servus* (Say), Et = *Euschistus tristigmus* (Say), Nv = *Nezara viridula* (L.), Op = *Oebalus pugnax* (F.), Pg = *Piezodorus guildinii* (Westwood). ^2^ Stink bug stage abbreviations: E = egg, N = nymph, A = adult. ^3^ Lab evaluation only.

### 1.4. Integration of Stink Bugs’ Natural Enemies into IPM Programs

The broad suite of natural enemies of stink bugs offers potential for managing these pests on a larger spatial scale, as their natural enemies may have the capacity to move across the landscape to affect stink bug populations in a range of habitats. It should be noted, however, that nearly all of the natural enemies noted in Table 1, Table 2 and Table 3 are known to attack multiple species of stink bugs and, in many cases, prey/host species from other families and orders. These generalized feeding and host relationships among the natural enemies can complicate the outcomes of biological control. Nevertheless, the breadth of host plant and habitat relationships for natural enemies of stink bugs, along with the diverse complex of natural enemy species and guilds, suggests that natural enemies have great potential to significantly affect stink bug populations.

There are several considerations for the biological control of stink bugs. First, a given crop system may be attacked by a complex of stink bugs rather than a single species (e.g., [54,75], and there are often crop-specific variations in the communities and activity of natural enemies against the respective stink bugs (e.g., [79,187,214]). These differences may yield highly variable outcomes depending on the stink bug and the natural enemy species making up the respective complexes in the crop.

Second, the life cycle of the stink bugs in the crop is important for biological control considerations. For example, pest stink bugs readily reproduce and develop populations in soybean and maize crops [75]. Thus, all life stages of the bugs can be found in the crop, and egg stages and young nymphs can be targeted with parasitoids and generalist predators, respectively, to slow population growth. In contrast, in temperate cotton production, stink bugs do their most serious damage largely as immigrant adults attacking the fruiting structures of the plant late in the season [215,216]; they reproduce relatively little in the crop and migrate into the fields as the cotton is in a susceptible stage. They tend not to build up their populations via reproduction within the cotton crop preceding their attack on bolls. Furthermore, the window of the cotton crop’s susceptibility to stink bug damage is sufficiently short that slowing population growth in the crop is not a viable option for limiting damage, nor would reliance on adult parasitoids provide a sufficiently rapid reduction in stink bug adults to significantly reduce the damage. Related to this is the importance of the relative timing of the mortality in the stink bug’s life for biological control. Liljesthröm and Bernstein [217] noted that stink bug egg parasitism was an important mortality factor within generations, whereas adult parasitism was an important factor between generations. This becomes very important in designing single-field or area-wide biological control plans, where a single generation (or less) may be involved at the field level but multiple stink bug generations are relevant over a larger area with multiple crops or crop phenologies.

#### 1.4.1. Classical/Importation Biological Control

The most widely applied form of biological control with stink bugs to date has been the classical/importation biological control of invasive stink bug species using parasitoids [218]. For example, the introduction of the exotic parasitoid *T. basalis* has provided at least partial success against *N. viridula* [43] (see also [214,219]), although *N. viridula* continues as an economic problem throughout most of its invaded range (e.g., [77,219,220]). *T. basalis* is established throughout the range of *N. viridula* in the Americas, and parasitism rates by this parasitoid can be quite high. However, parasitism by this parasitoid is also highly variable in space and time, and it is unclear how much reduction in the *N. viridula* population the parasitoids are actually causing [220]. This parasitoid also attacks other native pest stink bugs (Table 1), as well as the eggs of non-pest species (e.g., [221]), although the extent of its non-target impact has not been evaluated. Similarly, the adventive egg parasitoid *T. japonicus* was recently established in North America [44] and may reduce populations of the exotic BMSB. Introductions of appropriate exotic natural enemies will most likely continue to play a prominent role in managing invasive stink bug pests.

#### 1.4.2. Conservation Biological Control

Conservation biological control offers opportunities to actively reduce stink bug populations as a component of broader IPM programs, given the diverse complex of natural enemies (Table 1, Table 2 and Table 3). For example, Jones [214] demonstrated the value of the predator complex in macadamia for the control of *N. viridula*. However, most intentional conservation biological control requires considerable information on the biology of the natural enemies, including their needed resources, along with the biology of the pest, in relation to the ecology of the cropping system and its surrounding environment. Natural enemy conservation at its simplest for most conventional cropping systems may come in the form of reducing/modifying insecticide use to protect extant enemy populations, such as utilizing less-disruptive insecticides where available and/or localizing treatments to field borders or trap-crop areas (e.g., [49]) as colonization is occurring. The paucity of more stink-bug-specific insecticides limits options for directly integrating insecticides with biological control of stink bugs, but spatial or temporal separation of broad-spectrum insecticides and natural enemies to the greatest extent possible would be beneficial.

Given the potentially diverse complex of natural enemies of stink bugs, habitat manipulation/diversification to encourage natural enemies is an area of high potential e.g., [222], but there has been limited successful application. Stink bug parasitoids and predators are known to utilize and benefit from plant-derived resources (e.g., [119,160,168]) and concentrating stink bugs in trap patches adjacent to resource habitats for natural enemies may foster biological control of developing stink bug populations. Conservation approaches would be most useful in systems in which stink bugs reproduce and pass through at least one full generation, so that the full suite of available natural enemies can be brought to bear across all life stages and hinder population growth. More information is critically needed on habitat resource requirements for stink bugs’ natural enemies, especially relative to pest benefits and how best to deploy those resources in locally adapted approaches. This is especially true for field implementation of lab results and field-based assessments of the effects of habitat manipulation on stink bug populations and crop injury. In the absence of this information, efforts will continue to be haphazard, lacking an intentional and systematically efficacious structure.

#### 1.4.3. Augmentative Biological Control

Augmentation of natural enemies has not been utilized to any significant degree in North America for managing stink bug pests. Indeed, there are no stink-bug-specific natural enemies commercially available for such usage. In contrast, an effective augmentative program using egg parasitoids against *N. viridula* and *Euschistus heros* (F.) was used in Brazilian soybeans from the 1990s through to 2010 in a comprehensive IPM program. The parasitoid *T. basalis* was mass reared and released inundatively over large areas in Brazil when soybeans were flowering and stink bugs were initially colonizing the crop [153]. Caterpillar pests were largely controlled by applications of nucleopolyhedrosis virus, minimizing disruptions of natural enemies by pesticides applied against caterpillars. Unfortunately the program was abandoned when cropping practices changed (i.e., no-till cropping that required more insecticide use to treat soil-dwelling insect pests; intensive multicrop systems fostering year-round pest populations; increased use of fungicides to manage soybean rust) and rendered this approach unacceptable to producers as part of an overall IPM strategy [223]. Similar opportunities exist for developing effective augmentative programs in the US, but there has been a lack of concerted effort to develop such programs.

#### 1.4.4. Future Directions for Using Biological Control

Biological control approaches could significantly benefit from and add benefit when coupled with other tools, such as augmentative releases of natural enemies combined with the use of attract-and-kill tools, to enhance efficacy. For example, trap crops, such as sorghum. e.g., [49,222,224], may provide a tool to concentrate stink bug populations where predators and parasitoids might also be attracted by complementary resources, such as nectar and pollen in insectary plantings (e.g., [225]). Furthermore, parasitoids could be inoculatively released into early trap-crop plantings to reduce population growth in those concentrations early in the season as stink bug populations are building, and also to build natural enemy populations early in the season. Pheromones could be used to attract stink bugs [226] and their natural enemies [164,227], to serve as a nursery for natural enemies, and to slow stink bug population growth early in the season. Pheromones could also be used for monitoring colonizing stink bug populations [228,229] to better time parasitoid releases.

Abram et al. [220] utilized generalized stage-structured matrix models to simulate the impact of stink bug egg parasitism versus adult parasitism in relation to overall population growth for a generic stink bug (based on pooled life-history data from *N. viridula*, *H. halys*, and *B. hilaris*). The model indicated that egg parasitism may have little overall impact on stink bug population growth, as much/most of the egg parasitism causes replaceable mortality in the egg and nymphal stages. Adult parasitism, on the other hand, showed potential to significantly reduce population growth, contributing additive mortality. It was unclear how their model handled the modified fecundity and longevity resulting from adult parasitism. They further noted that the putatively limited effects of egg parasitism can be diminished even further if the overall mortality of the stink bugs is density-dependent. Liljesthröm and Bernstein [217] noted that mortality factors—i.e., egg parasitism, early nymphal (instars 1–3) mortality, and adult parasitism—all yielded apparently density-dependent effects on populations of *N. viridula* over 26 generations in Argentina, and noted significant mortality effects of both egg (within generations) and adult (across generations) parasitism. These studies suggest that integrating the complexes of natural enemies into intentional IPM systems may yield benefits for stink bug management. However, the population-level impact of natural enemies within and across life stages will likely vary with stink bug species and habitats.

#### 1.4.5. Area-Wide Biological Control Applications

Knipling [230] proposed that released natural enemies be used against target pest populations on larger geographic scales than individual fields in temperate regions. He suggested that releases should target the point(s) in the pest’s season when its populations are low (i.e., before reaching economic levels [231,232]) and when they may be spatially concentrated in defined areas, and/or late in the season with the intent to reduce the size of the overwintering population. Previous studies have investigated early-season predation when the enemy:pest ratio is highest [233,234]. Focusing on early-season predation for conservation or augmentative biological control may be the most economically viable approach, as the predators could maintain pests below economic levels [235]. Similarly, Knight and Gurr [76] suggested the potential of utilizing an area-wide approach to stink bug management as a viable option, integrating natural enemies with other practices, including trap crops.

Knipling [230] proposed identifying key incubator resources for pest populations early in the season while the pests are still limited in scope and scale, before there are extensive and diffuse resources available across the landscape to disperse the pest population. These population incubator locations can then be focal management areas to significantly slow incipient pest population growth. Knipling emphasized utilizing releases of appropriate natural enemies in combination with releases of sterile insects.

The success of Knipling’s approach with respect to natural enemies would require (1) a detailed knowledge of the spatial and temporal patterns of stink bug colonization and movement across the landscape in relation to host plant phenology, so that locales and time points could be anticipated for enemy releases [18]. It would also require (2) the identification of a natural enemy that would be appropriate for the pest species in question. In addition, (3) the capacity to rear and transport that natural enemy in sufficient numbers and at reasonable expense to justify and execute successful releases must be clear. (4) A monitoring network to assess the landscape would be necessary to track stink bugs in the landscape to target releases. This might be increasingly feasible with satellite imagery and artificial intelligence to pinpoint suitable host plant patches early in the season. In addition, (5) appropriate release ratios (enemy:pest) and methods would need to be determined. Beyond these biological and logistical elements, such a program would require (6) information exchange, agreement, and coordination across a regional group of producers with (7) sufficient resources and organization to execute and continuously evaluate the program and maintain accountability. Finally, (8) a clear demonstration of the economic and practical feasibility of undertaking such a program must be clearly demonstrated.

Such a large-scale program would likely require significant government oversight and coordination, with funding from various local and regional sources, similar to that in other large-scale programs, such as the boll weevil eradication program in the US [30,236]. Egg parasitoids lend themselves well to such an application, as noted above for soybeans in Brazil in a large-scale program similar to that outlined above. Appropriate pathogens also could be useful if they could be identified and efficiently propagated and released. Coupling area-wide management with local practices may yield very significant reductions in stink bug populations.

## 2. Conclusions

Stink bugs in the US have a large suite of natural enemies that offer opportunities for enhancing biological control efforts against these pests. The increasing regulatory losses of broad-spectrum insecticides, the serious environmental and health risks of these materials, and the decline in pesticide efficacy due to resistance development and variations among stink bug species and life stages all add pressure to the need to devise more sustainable management approaches for stink bug pests. Biological control offers considerable opportunities, but will require considerably more information about the ecology and behavior of natural enemies in the landscape, as well as more detailed assessments of their efficacy across a range of locations and cropping systems. These data can then be used to devise conservation and augmentation biological control systems that are appropriate for local needs. Local practices also would likely benefit from coordinated larger-scale approaches, given the high mobility and broad feeding ranges of stink bug pests.

## Data Availability

No new data were created or analyzed in this study. Data sharing is not applicable to this article.

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
