# Peer review of "Natural Enemies and Biological Control of Stink Bugs (Hemiptera: Heteroptera) in North America"

_insects, 2022, doi:10.3390/insects13100932_

Round 1

Reviewer 1 Report

This is a well-written and comprehensive review of the biological control of Stink Bugs. I don't have suggestions for the manuscript.

Author Response

Thank you!

Reviewer 2 Report

The authors have submitted a nice and, in general, well written review. I have only a few minor suggestions. 

First there is a little confusion in the studied geographic area. For example, some parts of the text, such as the titles of the tables and the conclusions (L449) refer to the United States, but the title and the tables refer to North America, including Canada and Mexico. Moreover, in table 3 and in some parts of the text, information from other countries, such as Cuba, Brazil and Argentina, is provided. I suggest reorganizing all this.

Please check the column “Distribution” in the tables as there are some incongruences such as the presence of some areas (e.g. Utah, Florida) and different ways to identify the areas (e.g. W US and Western US) 

L69: the common name of Bagrada hilaris is missing

LL169-170: the sentence seems to repeat a previously reported concept

LL202-207: I suggest reorganizing this part as the sentences seem disconnected and do not provide clear information

L238 ergatogyna instead of ergatogyn

L344-355: If it is possible, the topic of habitat management should be more stressed, for example by including the role of uncultivated areas or of companion plants.

L365: the sentence should be either better clarified or deleted.

Author Response

The authors have submitted a nice and, in general, well written review. I have only a few minor suggestions. 

RESPONSE: Thank you! We appreciate the very helpful comments and feedback.

 First there is a little confusion in the studied geographic area. For example, some parts of the text, such as the titles of the tables and the conclusions (L449) refer to the United States, but the title and the tables refer to North America, including Canada and Mexico. Moreover, in table 3 and in some parts of the text, information from other countries, such as Cuba, Brazil and Argentina, is provided. I suggest reorganizing all this.

RESPONSE: Thank you for catching this discrepancy. We had vacillated between focusing on the US and North America, and there were some relics of that vacillation in the text. We revised the table headings to “North America”. References to Brazil and Argentina are just referring to examples of enemy biology or pest management approaches observed elsewhere. Just an aside, Cuba is part of North America.

Please check the column “Distribution” in the tables as there are some incongruences such as the presence of some areas (e.g. Utah, Florida) and different ways to identify the areas (e.g. W US and Western US) 

RESPONSE: Thanks for catching this! We have standardized the identifications and the style in all three tables.

L69: the common name of Bagrada hilaris is missing

RESPONSE: Nice catch. Thank you! We have added the common name here.

LL169-170: the sentence seems to repeat a previously reported concept

RESPONSE: We modified Lines 164-169 to make the point more clear (that the parasitoid has only been recovered from kudzu bugs so far in North America) and less redundant.

LL202-207: I suggest reorganizing this part as the sentences seem disconnected and do not provide clear information

RESPONSE: We edited this portion to improve flow and clarity

L238 ergatogyna instead of ergatogyn

RESPONSE: Fixed

L344-355: If it is possible, the topic of habitat management should be more stressed, for example by including the role of uncultivated areas or of companion plants.

RESPONSE: We agree that this is an important area for biological control, and have addressed elements of it throughout the manuscript. Unfortunately, it remains a potential area of opportunity, with very little evidence for actual benefit. Much more work is needed, and this is lacking as we point out in the manuscript. For these reasons, we feel it best to add nothing further on the topic so as not to further increase the amount of unsupported speculation.

L365: the sentence should be either better clarified or deleted.

RESPONSE: We fleshed it out a bit to better clarify the point. Hopefully this makes more sense now.

Reviewer 3 Report

In this article, the authors provide a brief, but comprehensive, review on the natural enemies of the Stink bug in NA. I think that the paper is well written, and worthy of being published. I only have few minor comments listed below.

Lines 44-46: I would suggest to better describe how Stink Bug feed, mentioning the 4 different feeding habits. In fact, they can suck other fluid from cells or phloem.

Line 162: even if the ant predation is reported in the table 2, I would suggest to include in this paragraph few lines to describe this predatory activity, specially considering their impact on invasi species (see Castracani, Cristina, et al. "Predatory ability of the ant Crematogaster scutellaris on the brown marmorated stink bug Halyomorpha halys." Journal of Pest Science 90.4 (2017): 1181-1190.)

Line 193: it is well-known that BMSB is referred to the brown marmorated stink bug, but it should be said somewhere.

Lines 352-354: I would say that some data about that are available, even if not strictly related to NA (see Foti, Maria Cristina, et al. "Chemical ecology meets conservation biological control: identifying plant volatiles as predictors of floral resource suitability for an egg parasitoid of stink bugs." Journal of Pest Science 90.1 (2017): 299-310.)

Tables 1, 2 and 3: the area identified in the table captions (US, but I suppose it was intended NA) is not consistent with the countries listed inside the tables (e.g. Cuba, Brazil and Argentina)

Author Response

In this article, the authors provide a brief, but comprehensive, review on the natural enemies of the Stink bug in NA. I think that the paper is well written, and worthy of being published. I only have few minor comments listed below.

RESPONSE: Thank you! We appreciate the very helpful comments and feedback.

Lines 44-46: I would suggest to better describe how Stink Bug feed, mentioning the 4 different feeding habits. In fact, they can suck other fluid from cells or phloem.

RESPONSE: As this manuscript is not focused on stink bugs, per se, we feel that it would be inappropriate to add more significant details about the feeding ecology/behavior of the stink bugs beyond what is presented. We do agree that stating that they feed on phoem fluids is to restrictive, and have removed the word “phloem from the sentence. It now reads “Adults and nymphs suck [omitted – ‘phloem’] fluids from various plant parts such as…”

Line 162: even if the ant predation is reported in the table 2, I would suggest to include in this paragraph few lines to describe this predatory activity, specially considering their impact on invasi species (see Castracani, Cristina, et al. "Predatory ability of the ant Crematogaster scutellaris on the brown marmorated stink bug Halyomorpha halys." Journal of Pest Science 90.4 (2017): 1181-1190.)

RESPONSE: We inserted a sentence in this section noting that ants and tettigoniids were relatively more important in two different cropping systems – “For example, the highest rates of predation on sentinel eggs of N. viridula in soybeans and peanuts were attributed to tettigoniids and red imported fire ants, Solenopsis invicta, respectively, in a replicated large-plot study [182]. Thus, habitat/crop type influences predator presence and relative activity.” Although ants can be very important (and are especially impressive in lab studies as that referenced by the reviewer), they are one piece of the overall puzzle, and vary in activity across systems, as noted in the inserted study.

Line 193: it is well-known that BMSB is referred to the brown marmorated stink bug, but it should be said somewhere.

RESPONSE: This usage of the acronym BMSB for Halyomorpha halys was initially noted in Line 67 of the original manuscript.

Lines 352-354: I would say that some data about that are available, even if not strictly related to NA (see Foti, Maria Cristina, et al. "Chemical ecology meets conservation biological control: identifying plant volatiles as predictors of floral resource suitability for an egg parasitoid of stink bugs." Journal of Pest Science 90.1 (2017): 299-310.)

RESPONSE: Some data are available, and these references are cited in the text of the manuscript (we added the reference cited in the suggestion elsewhere in the manuscript. Thank you!). The problem with this paper and others like it is that they are conducted in the lab with no field validation of results. In the absence of actual field results, the work remains interesting speculation, and we stand by our position that more information is critically needed. We added an additional sentence here to clarify that field studies are especially needed – “This is especially true for field implementation of lab results, and field-based assessments of the effects of habitat manipulation on stink bug populations and crop injury.”

Tables 1, 2 and 3: the area identified in the table captions (US, but I suppose it was intended NA) is not consistent with the countries listed inside the tables (e.g. Cuba, Brazil and Argentina)

RESPONSE: Thank you! We have relabeled the table headings and cleaned up the location reporting in the tables.